# Improving the Assimilation of Enhanced Atmospheric Motion Vectors for Hurricane Intensity Predictions with HWRF

**Xu Lu *** , **Benjamin Davis** and **Xuguang Wang**

School of Meteorology, University of Oklahoma, Norman, OK 73072, USA; benjamin.j.davis-1@ou.edu (B.D.); xuguang.wang@ou.edu (X.W.)
\* Correspondence: luxu@ou.edu

**Abstract:** The initial conditions for hurricanes are difficult to improve due to the lack of inner-core observations over the ocean. An enhanced atmospheric motion vectors (AMVs) dataset from the Cooperative Institute for Meteorological Satellite Studies (CIMSS) has recently become available and covers the inner-core region of hurricanes. This study tries to find an optimal data assimilation (DA) configuration to better utilize the observations for the Hurricane Weather Research and Forecasting (HWRF) model with hurricane Irma (2017). The results show that (a) without vortex relocation (VR), the hourly three-dimensional ensemble–variational (3DEnVar) outperforms the 6-hourly 3DEnVar DA configuration in almost all aspects, except for long-term track predictions. The assimilation of inner-core AMVs further improves the corresponding intensity forecasts for both hourly and 6-hourly 3DEnVar DA. (b) The 6-hourly 3DEnVar DA predictions with VR can be significantly improved upon their non-VR counterparts. However, VR can be detrimental to hourly 3DEnVar minimum sea level pressure (MSLP) predictions due to the spuriously enhanced upper-level warm core. The improvements from the assimilation of additional inner-core AMVs are thus limited under hourly VR. Reducing VR frequency can reduce the detrimental effects of hourly 3DEnVar. (c) An updated observation error profile for the enhanced AMVs benefits the hourly 3DEnVar DA more than the 6-hourly 3DEnVar DA.

**Keywords:** data assimilation; enhanced atmospheric motion vectors; hurricanes; hourly 3DEnVar; HWRF

## 1. Introduction

One known difficulty in hurricane predictions is how to obtain accurate initial conditions for the numerical weather prediction (NWP) models over the open ocean. Recent advances in the assimilation of airborne-based inner-core observations have shown promising results in improving hurricane predictions. For example, Wick et al. (2018) [1] and Christophersen et al. (2018) [2] showed that the assimilation of dropwindsondes from the Global Hawk unmanned aircraft can improve the position and minimum sea level pressure (MSLP) analyses and predictions of tropical cyclones (TCs), especially for the non-steady-state storms. Lu and Wang (2020) [3] and Feng and Wang (2019) [4] demonstrated significant improvements in the storm structure analysis and predictions after assimilating the high-definition sounding system (HDSS) and expendable digital dropsondes (XDD) [5] deployed from the WB-57 [6]. The assimilation of tail Doppler radar (TDR) observations and high-density observations (HDOB) sampled onboard the National Oceanic and Atmospheric Administration (NOAA) P3 and G-IV aircrafts was also found critical in improving the TC initial conditions and the subsequent intensity predictions through extensive studies [3,7–14].

However, those airborne observations are often limited by the nature of the aircraft, which has restricted airtime and can only be launched from the coastal regions with strict constraints in range and frequency. The temporal and spatial discontinuity of the airborne

observations requires supplementary observations to fill in the gaps. While the direct assimilation of all-sky radiance observations is still immature [15–17], assimilating the satellite-derived atmospheric motion vectors (AMVs) is one of the alternatives to better utilize the satellite observations [18–21].

AMVs are derived from the movements of coherent water vapor targets between sequential satellite images to retrieve wind and height information of the atmosphere [21]. Such retrieval algorithms for AMVs are constantly evolving [22]. With the efforts of the Cooperative Institute for Meteorological Satellite Studies (CIMSS) at the University of Wisconsin, an enhanced AMV dataset [20,23,24] has recently become available with larger coverage, better quality, and higher density than the AMVs that were used previously in operational models, such as the Hurricane Weather Research and Forecasting (HWRF) model [20]. One of the advantages of this new dataset is the high-density coverage above the hurricane inner-core regions, which was not available before. In recent studies, this enhanced AMV dataset was found to help improve hurricane predictions in multiple aspects [20,23–25]. For instance, both Wu et al. (2014, 2015) [23,24] and Zhang et al. (2018) [25] showed that the assimilation of enhanced AMVs, especially those above the inner-core region, reduces the structure, track, and intensity errors of TC predictions. Based on the operational Hurricane Weather Research and Forecasting (HWRF) model framework, the preliminary research was performed using a 6-hourly three-dimensional ensemble–variational (3DEnVar) data assimilation (DA) method, even though the dataset itself is at an hourly frequency or even higher (e.g., 15 min in rapid scan mode). As suggested by Velden et al. (2017) [20], the high temporal density of the enhanced AMVs may not be efficiently utilized in the 6-hourly 3DEnVar DA. Early studies with high-resolution inner-core TDR observations [9,26] suggested the 6-hourly four-dimensional ensemble–variational (4DEnVar) or hourly 3DEnVar DA methods, which consider the temporal evolution of error covariance, can both outperform the 6 h 3DEnVar with the assimilation of those observations with high temporal density for hurricane predictions. This study is among the first to explore the optimal configuration of the enhanced AMV assimilation with hourly frequency DA using a newly developed GSI-based, continuously cycled, dual-resolution, hybrid ensemble–variational (EnVar) DA system for the HWRF model [8,9]. Further, the importance of inner-core AMVs in such DA configurations is also investigated.

Since the early years of hurricane NWP, vortex initialization (VI) techniques have been used to provide location and intensity corrections over open water [27–31]. The VI technique used in the operational HWRF was developed by Liu et al. (2006, 2000) [32,33]. It primarily contains two steps: vortex relocation (VR), which corrects the storm location; and vortex modification (VM), which modifies the storm size and intensity [34]. Previous studies with HWRF showed that the VI, especially the VR component, is necessary for a continuously cycled DA system in hurricane predictions when no continuous inner-core observations are available [9]. While the enhanced AMVs are continuously available and cover the hurricane inner-core regions [24,25], studies by Velden et al. (2017) [20] and Zhang et al. (2018) [25] suggested that the benefit of assimilating the enhanced AMVs may be reduced when using VI. Therefore, it is necessary to investigate whether the VR is still needed in a fully cycling DA system with the assimilation of the enhanced AMVs.

As indicated by Velden et al. (2017) [20], quality control (QC), such as the observation error profile and gross error check employed for this newly developed AMV dataset, affects the performance of DA and the subsequent forecasts. The observation error profile is further updated in response to the enhanced AMVs' recent updates (Velden, C., personal communication, 2020). Therefore, this study further investigates the impact of this updated error profile in the advanced 3DEnVar DA system for HWRF.

This paper is organized as follows. The model, case, observations, and experiment designs are described in Section 2. Section 3 discusses the results of the experiments. Conclusions and further discussions are included in Section 4.

## 2. System, Model, Observations, and Experiment Design

### 2.1. System and Model Description

The Gridpoint Statistical Interpolation (GSI)-based hybrid EnVar DA system for HWRF was developed by Lu et al. (2017) [8,9]. Following the 2017 operational HWRF model configurations [34], the model was further adapted and upgraded with hourly DA capabilities [26]. Specifically, this system includes a 40-member self-cycled HWRF ensemble Kalman filter (EnKF) running in parallel with a deterministic EnVar. The EnKF ensemble is running at a coarser resolution than the deterministic EnVar. Both the lateral boundary conditions and initial conditions are provided by the operational global forecast system (GFS) [35]. Before DA, VR will be applied to the ensemble and deterministic forecasts following Liu et al. (2000, 2006) [32,33] and Lu et al. (2017a,b) [8,9] based on the demand of experiment designs (Section 2.4). In the DA step, the coarse resolution ensemble provides the full ensemble error covariance for the high-resolution deterministic EnVar through an augmented control vector (GSI-ACV) method [35–37]. Meanwhile, an ensemble square root Kalman filter (EnSRF) method [38] is applied to the ensemble utilizing the observation preprocessing and forward operators from GSI. As a two-way system, the ensemble analysis is recentered (replacing the ensemble mean) with the EnVar analysis (Wang et al. 2013, Figure 1) [35]. Then, both the recentered EnKF analysis and the EnVar analysis are used to initialize a 6- or 1 h forecast to prepare for the next DA cycle, depending on the need for a 6-hourly or hourly 3DEnVar. No ocean coupling is applied during the forecasts. More details about this hybrid EnVar DA system and the methodology for 3DEnVar can be found in Lu et al. (2017) [8,9].

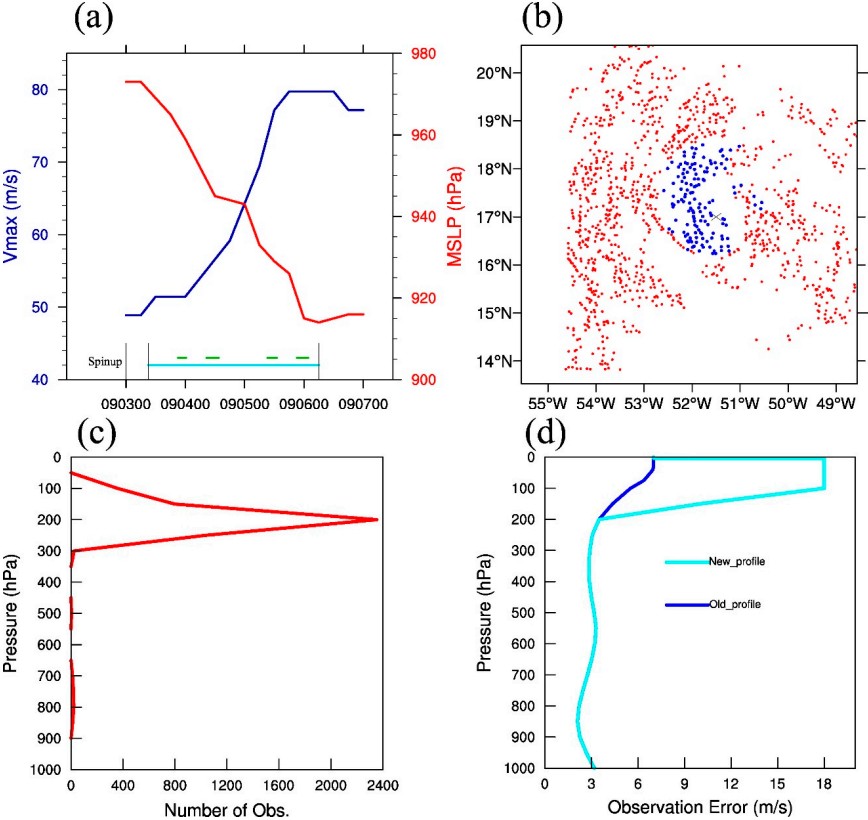

**Figure 1.** (**a**) Vmax (blue) and MSLP (red) evolution of hurricane Irma (2017). The availability of TDR observations and the enhanced AMVs are shown in green and cyan lines, respectively. The (**b**) horizontal and (**c**) vertical distribution of the enhanced AMVs assimilated at 06:00 UTC on 4 September 2017. Blue dots in (**b**) are the observations above 125 hPa, and the red dots are below. The (**d**) default (blue) and updated (cyan) observation error profiles for the enhanced AMVs.

The HWRF model used in this study is HWRF version 3.9 [34]. The horizontal grid spacing is approximately 2 km (0.015°), 6 km (0.045°), and 18 km (0.135°) for the deterministic control for the inner, middle, and outer domains, respectively. The ensemble members are only running with 6 km and 18 km grid spacing to save computational resources. For both EnVar and EnKF, DA is only performed on the inner domains. Following the operational HWRF [34], the outermost domain will be updated with GFS analysis (or 3 h forecast at +3 synoptic times in hourly configuration) for initialization. The model tops at 10-hPa with 75 vertical levels. The model physics used in this study follows Biswas et al. (2018) [34] and is shown in Table 1.

**Table 1.** List of observations assimilated and model physics configurations.

| | Data/Scheme Details |
|---|---|
| Conventional Data Assimilated | Radiosondes; dropwindsondes; aircraft reports (aircraft report (AIREP)/pilot report (PIREP); reconnaissance code (RECCO), meteorological data collection and reporting system-aircraft communications addressing and reporting system (MDCRS-ACARS), tropospheric airborne meteorological data reporting (TAMDAR), aircraft meteorological data relay (AMDAR)); surface ship and buoy observations; surface observations over land; pibal winds; wind profilers; radar-derived velocity azimuth display (VAD) wind; WindSat scatterometer winds; integrated precipitable water derived from the Global Positioning System (GPS) |
| Satellite Radiance Data Assimilated | Infrared radiation (IR) instruments: high-resolution infrared radiation sounder (HIRS), atmospheric infrared sounder (AIRS), Infrared atmospheric sounding interferometer (IASI), geostationary operational environmental satellites (GOES) sounders, cross-track infrared sounder (CrIS), special sensor microwave imager/sounder (SSMIS); microwave (MW) instruments: advanced microwave sounding unit-A (AMSU-A), microwave humidity sounder (MHS), advanced technology microwave sounder (ATMS) |
| Other Data Assimilated | National Oceanic and Atmospheric Administration (NOAA) P3 tail Doppler radar (TDR); hurricane and severe storm sentinel (HS3) Global Hawk (GH) dropsonde; tropical cyclone (TC) vital mean sea level pressure (MSLP); high-density flight-level wind, temperature, and moisture observations; Enhanced Cooperative Institute for Meteorological Satellite Studies (CIMSS) atmospheric motion vectors (AMV) |
| Physics Scheme Used | Ferrier–Aligo microphysics scheme with minor updates [39]; scale-aware simplified Arakawa–Schubert (SASAS) cumulus scheme [40–43]; Hurricane Weather Research and Forecasting (HWRF) modified Geophysical Fluid Dynamics Laboratory (GFDL) surface-layer scheme [44–46]; Noah land-surface model [47,48]; non-local hybrid eddy-diffusivity mass-flux (hybrid EDMF) planetary boundary layer (PBL) scheme [49–51]; Rapid Radiative Transfer Model for Global Circulation Models (RRTMG) longwave and shortwave radiation schemes [52,53] |

### 2.2. Case Description

Hurricane Irma (2017) is a category 5 hurricane that formed as a tropical depression at 00:00 UTC on 30 August 2017 and dissipated at 18:00 UTC on 13 September 2017. Irma first hit Barbuda around 05:45 UTC on 6 September as a category 5 hurricane and fluctuated between category 5 and 4 within the next three days with its additional four landfalls along the Caribbean Islands. Passing through Cudjoe Key in the Florida Keys at category 4, it made the final landfall near Marco Island, Florida, at 19:30 UTC on 10 September as a category 3 hurricane. This two-week-long TC caused more than 100 direct and indirect deaths along its seven landfalls [54]. As indicated in Cangialosi et al. (2017) [54], a major difficulty for Irma prediction was its rapid intensification (RI) and the landfall uncertainties (eastern or western Florida) in its early stages. Such challenges make Irma a good case to explore the impact of the optimal configurations for the enhanced AMVs, which are expected to provide both environmental information that improves the track forecasts and inner-core information that improves the intensity predictions.

### 2.3. Observations and Preprocessing

The observations used in this study were listed in Table 1. Generally, this study assimilates all the observations used in the operational HWRF during hurricane Irma (2017), except for replacing the operational AMVs with the enhanced AMVs from CIMSS. The enhanced AMVs are available at an hourly frequency from 09:00 UTC, 3 September 2017 to 06:00 UTC, 6 September 2017 (Figure 1a). As stated in the Introduction, Irma grew to a category 5 hurricane from category 2 and made its first landfall in Barbuda during the period.

Following the criteria from Velden et al. (2017) and Wu et al. (2014, 2015) [20,23,24], the enhanced AMVs are additionally preprocessed before DA. To be specific, only the observations with a quality indicator (QI) equal to or greater than 0.8 are used in this study. The quality-controlled observations are then superobbed (or averaged) within 0.055- by 0.055-degree (~6 km) boxes. Figure 1b shows an example of the horizontal distribution of the preprocessed AMV observations valid at 06:00 UTC on 4 September 2017. By default, the AMVs above the 125 hPa will be discarded by GSI. The observations that are above 125 hPa can be considered as from the inner-core regions (with deep convection) and are marked in blue dots. The vertical distribution of the same preprocessed AMVs from the innermost domain (2 km) is shown in Figure 1c. It clearly shows that the enhanced observations for this innermost domain peaked at around 200 hPa. The vertical profile of the default observation error profile used by the operational HWRF is shown in Figure 1d along with the modified error profile. The major difference is the enlarged observation errors in the upper levels above 200 hPa (Chris Velden, personal communication, 2020). Such an increase in the upper levels is due to having more observations to better estimate the root mean square error (RMSE) for those levels in the newer dataset than before.

### 2.4. Experiment Design

To address the scientific goals of this study, twelve sets of experiments have been designed, as shown in Table 2. They are named as "6H_NVR", "6H_NVR_N125", "6H_VR", "6H_VR_N125", "6H_NP", "1H_NVR", "1H_NVR_N125", "1H_VR1", "1H_VR1_N125", "1H_VR6", "1H_VR6_N125", "1H_NP", respectively. Each experiment is detailed as follows.

**Table 2.** List of experiments.

| Experiment Name\Features | VR | VR Frequency | DA Frequency | Inner-Core AMV (Above 125 hPa) | Error Profile |
|---|---|---|---|---|---|
| 6H_NVR | N | None | 6-Hourly | Y | Default |
| 6H_NVR_N125 | | | | N | |
| 6H_VR | Y | 6-Hourly | | Y | |
| 6H_VR_N125 | | | | N | |
| 6H_NP | | | | Y | Updated |
| 1H_NVR | N | None | Hourly | Y | Default |
| 1H_NVR_N125 | | | | N | |
| 1H_VR1 | Y | Hourly | | Y | |
| 1H_VR1_N125 | | | | N | |
| 1H_VR6 | | 6-Hourly | | Y | |
| 1H_VR6_N125 | | | | N | |
| 1H_NP | | | | Y | Updated |

Experiment "6H_NVR" is the baseline experiment for the six-hourly 3DEnVar DA configuration. It is initialized at 00:00 UTC on 3 September 2017 from the GFS and cycled every 6 h, assimilating the observations in Table 1 until 06:00 UTC on 6 September 2017.

Note, the enhanced AMVs are available since 12:00 UTC, 3 September 2017, and we only assimilate the closest hour of AMVs due to the data density. There are, in total, 11 DA cycles that are counted for the AMV assimilation. No VR is performed on either the ensemble or deterministic background.

Experiment "6H_NVR_N125" differs from experiment "6H_NVR" by excluding the AMV observations above 125 hPa. The goal of the comparison between "6H_NVR_N125" and "6H_NVR" is to investigate the impact of inner-core AMVs.

Different from "6H_NVR" ("6H_NVR_N125"), experiment "6H_VR" ("6H_VR_N125") is conducted with VR before each DA cycle starting from 12:00 UTC, 3 September 2017, to understand the impact of VR under this 6-hourly 3DEnVar DA configuration. Additionally, the comparison between "6H_VR" and "6H_VR_N125" is made to further understand the impact of inner-core observation when VR is available.

The "6H_NP" experiment differs from "6H_VR" by using the updated error profile in Figure 1d to investigate the impact of such an error profile.

The "1H_NVR" and "1H_NVR_N125" experiments are comparable to their 6-hourly counterparts ("6H_NVR" and "6H_NVR_N125"), except using the hourly 3DEnVar DA configuration. They share the same spin-up background from 2017/09/03 00:00 UTC to 06:00 UTC, 3 September 2017, with the "6H" experiments. The first hourly 3DEn-Var DA cycle starts at 09:00 UTC, 3 September 2017, which is the 3 h forecast from 06:00 UTC, 3 September 2017. The comparison between "1H_NVR" ("1H_NVR_N125") and "6H_NVR" ("6H_NVR_N125") is made to explore the impact of the higher frequency DA of the enhanced AMVs when VR is not performed. Note, due to the design of the system and data availability, the data volume in the 6-hourly 3DEnVar at each synoptic time (00, 06, 12, 18 UTC) is not finished until 3 h later in the hourly 3DEnVar experiments. Therefore, for a fair comparison, we only consider the synoptic +3 h forecast from the hourly 3DEnVar experiments in this study, corresponding to the 6-hourly 3DEnVar forecast at the synoptic times.

To understand the impact of VR, experiments "1H_VR1" and "1H_VR6" are conducted to understand the optimal VR configuration for hourly 3DEnVar. By design, both "1H_VR1" and "1H_VR6" are based on "1H_NVR", except that "1H_VR1" performs VR every hour before DA, and "1H_VR6" only performs VR every 6 h prior to the DA at synoptic times. Further discussions on why and how the VR frequency matters are presented in Section 3.2. Comparisons between "1H_NVR", "1H_VR1" and "1H_VR6" show the impact of VR in the hourly 3DEnVar configuration when assimilating the enhanced AMVs. The corresponding experiments without assimilating inner-core observations, "1H_NVR_N125", "1H_VR1_N125", and "1H_VR6_N125", are also consistently performed to investigate the impact of inner-core observations in each scenario. Additionally, comparisons between "1H_VR6" ("1H_VR6_N125") and "6H_VR" ("6H_VR_N125") can present the impact of higher frequency DA of the enhanced AMVs when VR is performed.

Similar to "6H_NP" and "6H_VR", experiment "1H_NP" is performed based on "1H_VR6" by replacing the standard observation error profile with the updated one. This experiment is designed to explore the new error profile's impact on the enhanced AMVs' assimilation in the hourly 3DEnVar configuration.

## 3. Results

### 3.1. Impact of Inner-Core AMVs Assimilation without VR

As outlined in Section 2.4, this subsection first discusses the impact of the assimilation of the inner-core AMVs when no VR is performed in the continuously cycling 3DEnVar DA system with HWRF.

Figure 2a–c shows the RMSE of the eleven forecasts starting from 12:00 UTC, 3 September 2017 to 00:00 UTC, 6 September 2017, for the 6-hourly 3DEnVar and the corresponding hourly 3DEnVar forecasts. The best track data are found in the Atlantic hurricane database (HURDAT2) from the National Hurricane Center (https://www.nhc.noaa.gov/data/hurdat/hurdat2-1851-2020-020922.txt, accessed on 8 April 2022). In the 6-hourly

3DEnVar DA configuration, while the track predictions are almost comparable (Figure 2c), the additional assimilation of inner-core AMVs ("6H_NVR") only slightly outperforms "6H_NVR_N125" in the early lead-time 10 m wind maximum (Vmax) and minimum sea level pressure (MSLP) predictions (Figure 2a,b). The mission-by-mission forecast in Figure 2d–i indicates multiple strong spin-downs (Vmax drop greater than 5 ms$^{-1}$ (6 h)$^{-1}$, [55]) in both experiments. "6H_NVR" appears to have slightly better resistance to the spin-down issue than "6H_NVR_N125" (Figure 2d vs. Figure 2g). To understand the issue, one of the spin-down cycles is further investigated below.

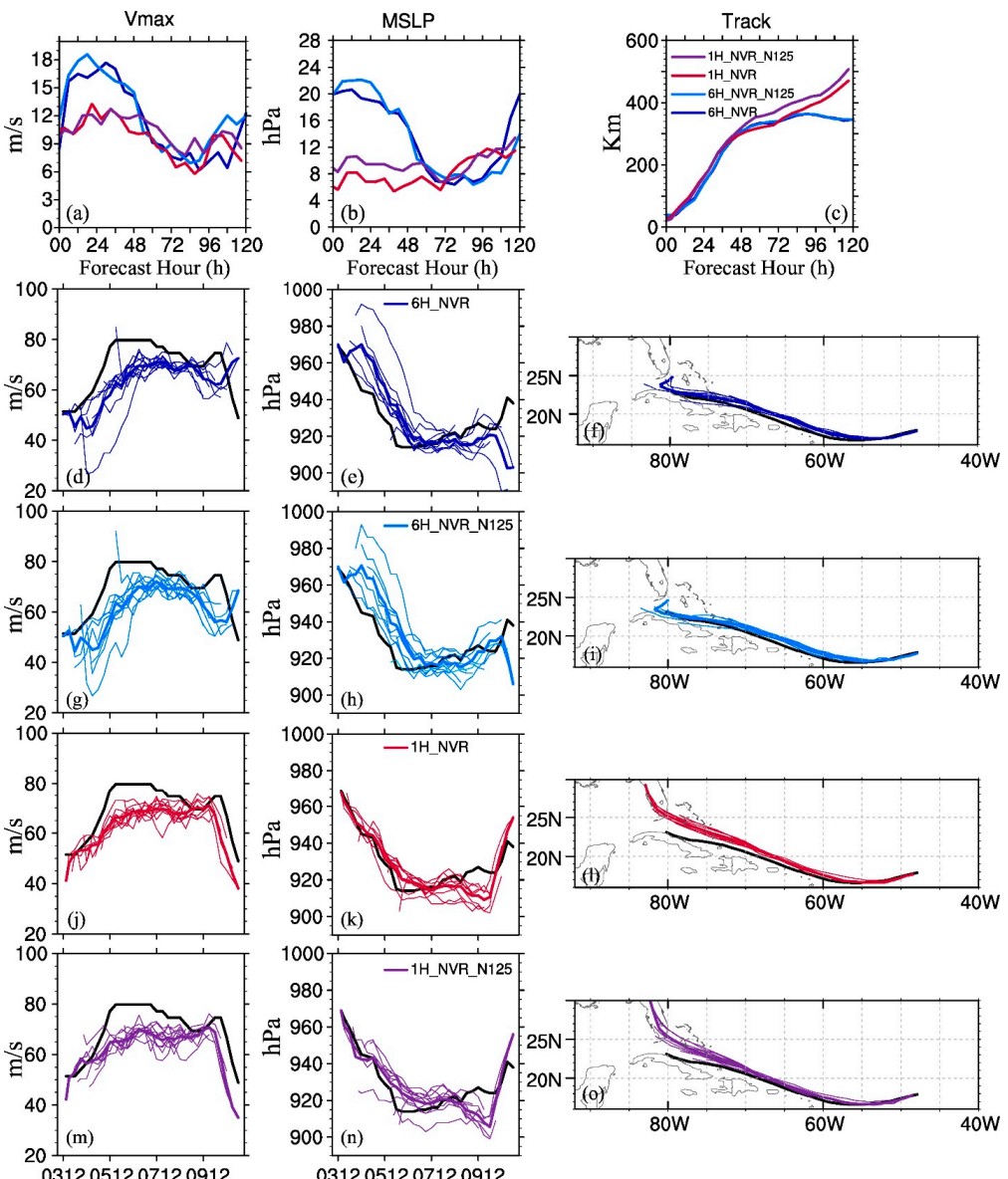

**Figure 2.** (**a**) Vmax, (**b**) MSLP, and (**c**) Track prediction RMSE for "1H_NVR" (red), "1H_NVR_N125" (purple), "6H_NVR" (blue), and "6H_NVR_N125" (turquoise). (**d–o**) show the individual forecast for each experiment for (**d,g,j**), and (**m**) Vmax, (**e,h,k,n**) MSLP, and (**f,i,l,o**) Track predictions and verification with the best track (black). The average of forecasts valid at each synoptic time is shown in bold color lines for each experiment for reference.

Figure 3 shows the 1 km height horizontal wind and south-to-north cross-section wind structures from the Hurricane Research Division (HRD) radar composite [56], and the analysis of each experiment is valid at 12:00 UTC, 4 September 2017. Fischer et al. (2020) [57] showed that Irma was going through an eyewall replacement cycle during this

RI period, and the observations in Figure 3f show features of the secondary eyewall around 60 km radii and an abnormal wind maximum aloft around 8 km height. As compared to the observations, the "6H_NVR" experiment failed to capture the double eyewall and produced a disorganized storm with a west–east elongated pattern. "6H_NVR_N125" looks to be even more disorganized than "6H_NVR" with more noisy pressure contours and an even larger eye size. Those large disagreements with the observations are found to be attributed to the large background location errors (they can be as large as 50 km for some cycles, not shown). As compared to "6H_NVR_N125", the slightly better "6H_NVR" suggests that the assimilation of the inner-core AMVs can still compensate for the location error to some extent during the cycling forecasts.

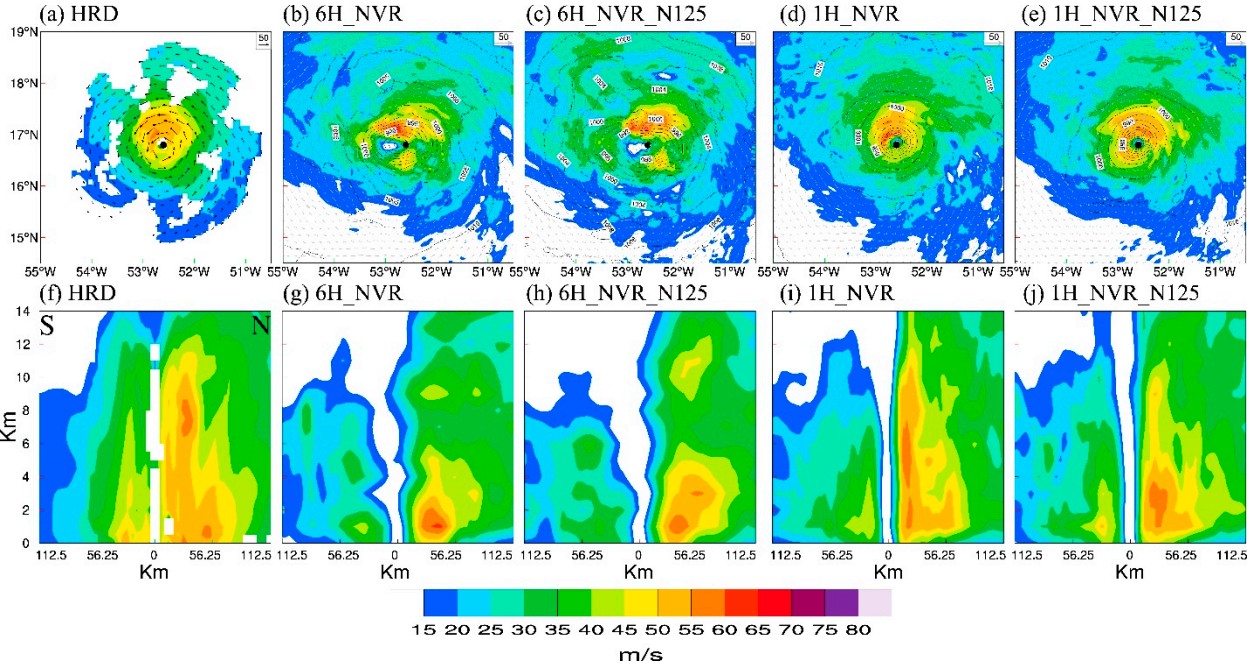

**Figure 3.** Horizontal wind (vector) and wind speed (shading) at (**a–e**) 1000 m height and (**f–j**) south-to-north cross-section for (**a,f**) the HRD radar composite, (**b,g**) "6H_NVR" analysis, (**c,h**) "6H_NVR_N125" analysis, (**d,i**) "1H_NVR" analysis, and (**e,j**) "1H_NVR_N125" analysis, valid at 12:00 UTC, 4 September 2017. The black dot in each horizontal figure indicates the best track location.

In comparison with the 6-hourly 3DEnVar experiments, the hourly 3DEnVar experiments perform generally better than their corresponding counterparts in almost all aspects, except for the long-term track predictions. For example, "1H_NVR_N125" produces significantly better Vmax and MSLP predictions than the "6H_NVR_N125" experiments at almost all cycles with much fewer spin-down issues (Figure 2a,b,m,n). Additionally, the "1H_NVR_N125" analysis structure in Figure 3,j fits the observations better with more circular wind patterns, a reduced eye size, and some weak features of the secondary eyewall as compared to "6H_NVR_N125". The northward long-term track prediction biases in hourly 3DEnVar experiments are found due to the initial condition differences in the outermost domain (not shown). To be specific, because of the design of the DA system in Section 2.1, the large-scale information from the outermost domain in hourly 3DEnVar forecasts is from the 3 h GFS forecasts. It is different from the 0 h GFS analysis, as used in the 6-hourly 3DEnVar experiment. In comparison with "1H_NVR_N125", the assimilation of additional inner-core AMVs in "1H_NVR" further improves the Vmax and MSLP predictions and the wind analysis patterns, especially the structures of the secondary eyewall and the secondary wind maximum aloft (Figures 2a–b and 3d,i). The track predictions in "1H_NVR" are also slightly better than "1H_NVR_N125".

Overall, the NVR experiments show that in both 6-hourly and hourly 3DEnVar configurations, the assimilation of additional inner-core AMVs can help slightly improve the Vmax and MSLP predictions by improving the structural analyses. Comparisons between the 6-hourly 3DEnVar experiments and their corresponding hourly 3DEnVar counterparts indicate the benefits of higher DA frequencies in assimilating the enhanced AMVs.

### 3.2. Impact of Inner-Core AMVs Assimilation with VR

Given that the background location error significantly degraded the analysis in "6H_NVR" and "6H_NVR_N125", this section explores the impact of assimilating inner-core AMVs with VR.

Like the current operational HWRF, experiments "6H_VR" and "6H_VR_N125" perform VR before each DA cycle every 6 h. This step reduces the background location error and was found necessary in early studies [9]. Consistently, the VR experiments significantly improve the Vmax and MSLP predictions, especially for the first 48–72 h compared to their corresponding NVR counterparts (Figure 4a,b). The analyzed structures in both "6H_VR" and "6H_VR_N125" are also more consistent with the observations, with more circular patterns, smoother pressure contour, tightened eye size, as well as the features of secondary eyewall, than "6H_NVR" and "6H_NVR_N125" (Figure 5 vs. Figure 3). Better consistency with the observations and best track again demonstrates the positive impact of VR for the 6-hourly 3DEnVar configurations.

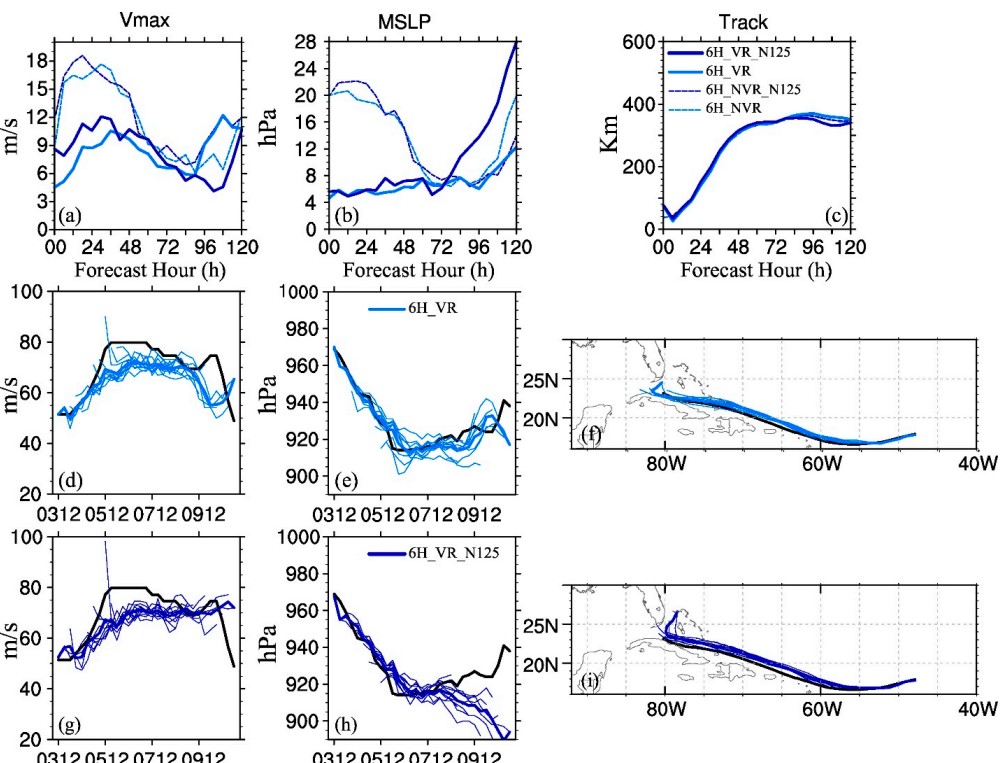

**Figure 4.** Same as Figure 2, except for "6H_VR" (solid turquoise), "6H_VR_N125" (solid blue), "6H_NVR" (dashed turquoise), and "6H_NVR_N125" (dashed blue).

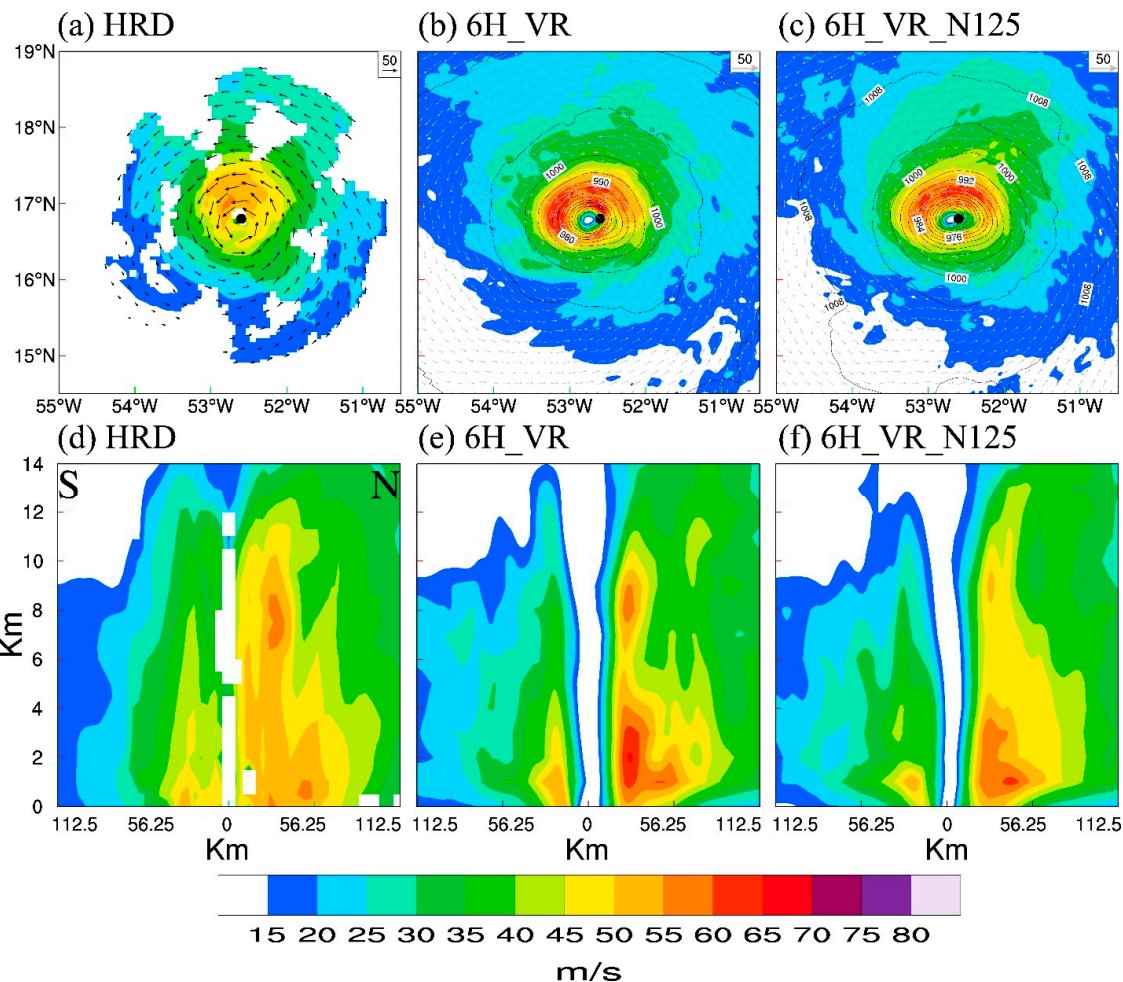

**Figure 5.** Same as Figure 3, except for (**b**,**e**) "6H_VR" and (**c**,**f**) "6H_VR_N125".

When "6H_VR" and "6H_VR_N125" are compared, the additional assimilation of inner-core AMVs in "6H_VR" outperforms "6H_VR_N125" in the analysis by better capturing the height of the aloft wind maximum and a clearer pattern of the double eyewall (Figure 5). Figure 6 shows the azimuthal mean radial wind analysis valid at 12:00 UTC, 5 September 2017, for "6H_VR" and "6H_VR_N125". At this stage of the storm, Irma was reaching its peak intensity and maintained category 5 intensity until it hit Barbuda about 18 h later (Figure 1a). Therefore, Irma should be a mature storm that fits the typical storm structure with a strong low-level inflow, as well as a strong and high upper-level outflow. "6H_VR" produces a slightly weaker but thicker upper-level outflow than "6H_VR_N125" (Figure 6a,b). Such a feature is due to the inner-core AMVs trying to pull the outflow maxima to a higher level, which shows a better match with the outflow observations around 100 hPa in both pattern and strength (Figure 7a,b). Consequently, "6H_VR" produces better Vmax and MSLP predictions than "6H_VR_N125" (Figure 4). The improvement includes alleviated spin-down issues, as shown in Figure 4d vs. Figure 4g. "6H_VR" also produces better long-term intensity predictions during the weakening stage than "6H_VR_N125" (Figure 4d,e,g,h). Nevertheless, the track predictions are overall comparable between the two experiments (Figure 4c,f,i).

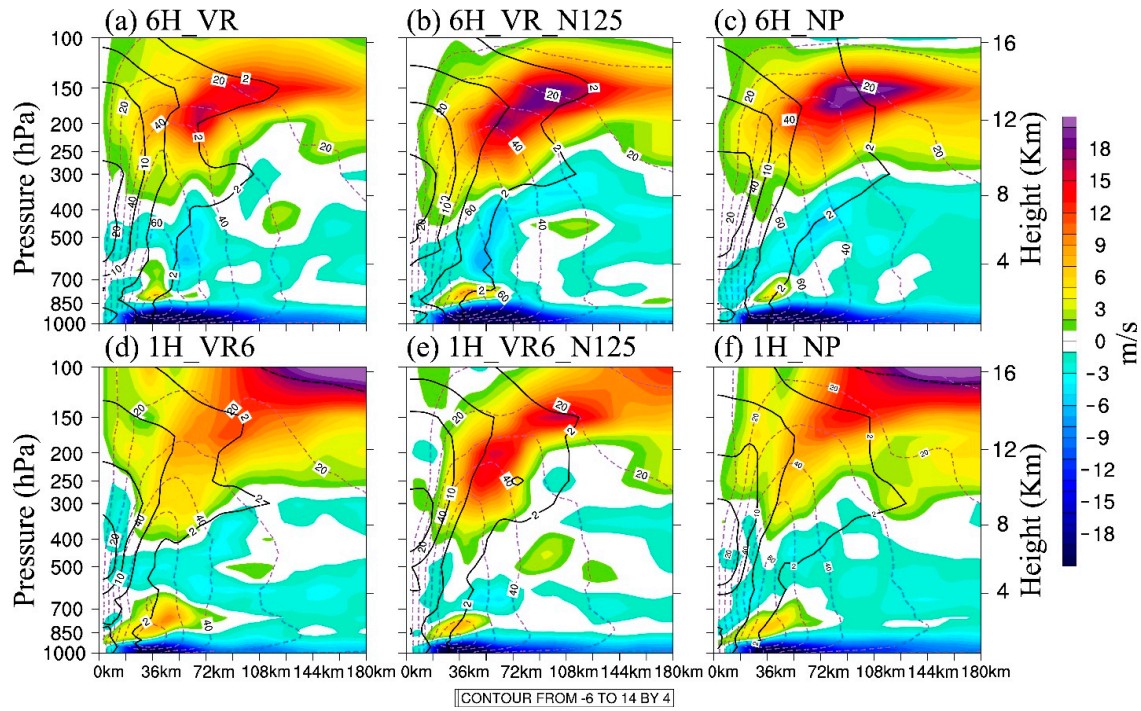

**Figure 6.** The azimuthal mean radial wind (shading), tangential wind (dashed contour), and temperature anomaly (solid contour) for (**a**) "6H_VR", (**b**) "6H_VR_N125", (**c**) "6H_NP", (**d**) "1H_VR6", (**e**) "1H_VR6_N125", and (**f**) "1H_NP" valid at 12:00 UTC, 5 September 2017.

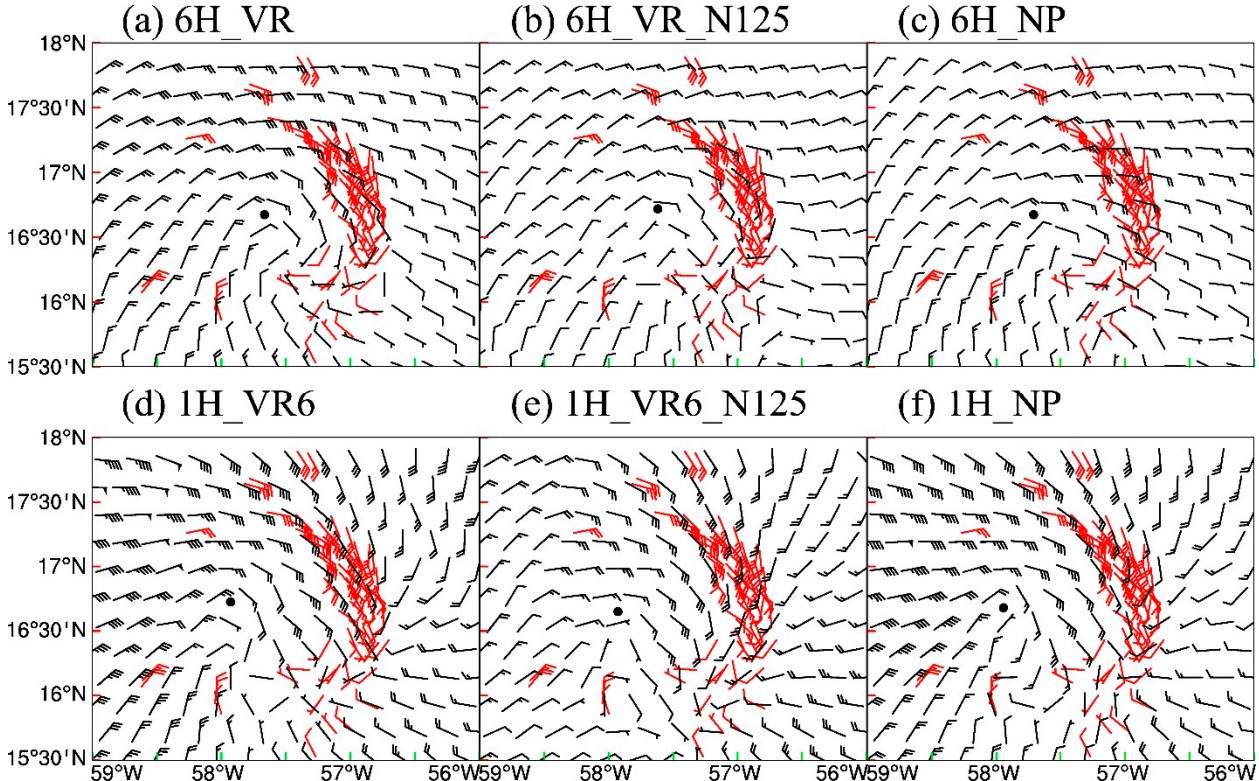

**Figure 7.** The 100 hPa horizontal wind flow (black) in verification with the AMVs (red; between 90hPa and 110 hPa) for (**a**) "6H_VR", (**b**) "6H_VR_N125", (**c**) "6H_NP", (**d**) "1H_VR6", (**e**) "1H_VR6_N125", and (**f**) "1H_NP" valid at 12:00 UTC, 5 September 2017. The black dot indicates the analysis storm center from the surface for each experiment.

While VR improves the 6-hourly 3DEnVar experiments, Figure 8 suggests that VR can bring negative impacts to the hourly 3DEnVar experiments. When performing VR before each DA cycle at an hourly frequency in "1H_VR1" and "1H_VR1_N125", the initial MSLP analysis becomes much worse than the corresponding 1H_NVR experiments (Figure 8b). Moreover, although Figure 8a indicates that the Vmax predictions in "1H_VR1" are improved upon "1H_VR1_N125" due to fewer spin-down issues (Figure 8d vs. Figure 8g), "1H_VR1" is producing even worse initial MSLP analyses than "1H_VR1_N125" (Figure 8e vs. Figure 8h). Figure 9 indicates that there is no significant issue in the wind pattern analyses as compared to either the observations (Figure 9a) or their 1H_NVR counterparts (Figure 3d,e,i,j). The clear patterns of the double eyewall and the middle-level secondary wind maximum in the north in "1H_VR1" are consistent with the observations. Further investigations into the 1H_VR1 experiments show that the degradation in the short-term MSLP predictions is likely attributed to the frequent VR. In the VI package from the operational HWRF [32,33], VR is performed by taking the background vortex out of the environment and then putting the vortex in the correct location. Figure 10a, b shows that during this VR step, even without moving the storm location, the temperature field will still be enhanced by about 3K at 9 km altitude during the out-and-back procedure. The increase in the temperature field in the upper-eye region then produces an artificially stronger upper-level warmer core as compared to the original background (Figure 10c vs. Figure 10d). Given there are no thermodynamic observations in the upper-level inner-core region available, the assimilation of dynamic observations, such as the TDR and upper-level AMVs, can only slightly reduce the warm core (Figure 10e,f vs. Figure 10c,d). Therefore, the temperature reduction from DA cannot cancel out the increase from relocation (Figure 10e vs. Figure 10f). While it takes time for the model to gradually rebalance the dynamic and thermodynamic fields, hourly VR in "1H_VR1" and "1H_VR1_N125" continuously enhances the spurious upper-level warm core and rapidly decreases the MSLP to an unreasonable value through the hydrostatic equilibrium.

To support the hypothesis, experiments "1H_VR6" and "1H_VR6_N125" are conducted to reduce the frequency of VR to once every 6 h, which is proved to be working for 6H_VR. Figure 11 shows that the reduction in VR frequency significantly improves the MSLP analysis and short-term predictions, as expected. However, in comparison with the NVR experiments "1H_NVR" and "1H_NVR_N125", experiments "1H_VR6" and "1H_VR6_N125" only slightly improve the MSLP predictions at the early lead times; the Vmax and Track predictions are generally comparable or slightly worse at longer lead times. Such a result is different from the VR impact for 6-hourly 3DEnVar, where the VR improves intensity predictions significantly, especially for the hourly DA of inner-core AMVs.

The intra-comparison between "1H_VR6" and "1H_VR6_N125" indicates that when VR exists, the inner-core AMVs from "1H_VR6" are slightly degrading the intensity predictions as compared to "1H_VR6_N125", especially for the long-term predictions after 72~80 h (Figure 11). This degradation from the assimilation of inner-core AMVs is conflicting with the hourly 3DEnVar experiments without VR ("1H_NVR" vs. "1H_NVR_N125"; Figure 2). Figure 12 shows that "1H_VR6_N125" is capturing the double eyewall and secondary wind maximum at 8 km height better than the corresponding "1H_VR6" analysis. The negative impacts from the imbalanced thermal and dynamical fields from VR likely dominate the inner-core region when the location error is not a major issue in such an hourly configuration.

However, further investigations into the secondary circulations indicate that the inner-core AMVs may still be necessary for a rapidly evolving hurricane. For example, Figure 6d indicates that the "1H_VR6" can produce a stronger inner-core structure with higher and stronger upper-level outflow as compared to "1H_VR6_N125" (Figure 6b). This stronger storm analysis is also found multiple times in Figure 11d. Figure 7d shows that the general outflow in "1H_VR6" around this level is stronger than "1H_VR6_N125" (Figure 7e), especially in the southern portion of the storm, where "1H_VR6" produces an outflow that is more consistent with the observations in both strength and directions than

"1H_VR6_N125". Quantitively, the RMSE against those observations from each analysis shows that "1H_VR6" (8.83 ms$^{-1}$) matches better than "1H_VR6_N125" (13.54 ms$^{-1}$).

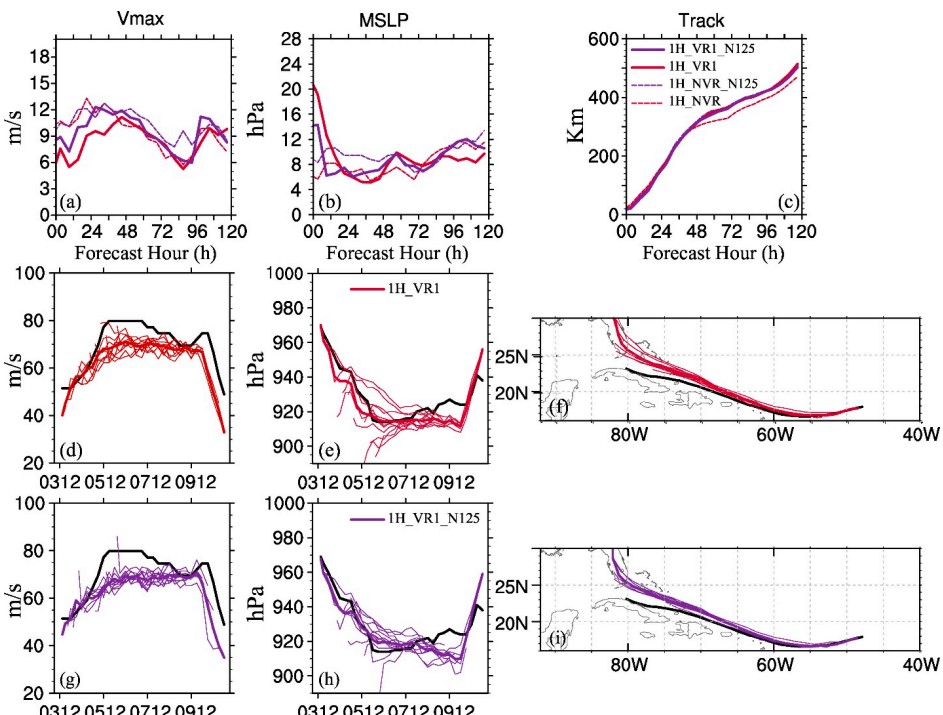

**Figure 8.** Same as Figure 2, except for "1H_VR1" (solid red), "1H_VR1_N125" (solid purple), "1H_NVR" (dashed red), and "1H_NVR_N125" (dashed purple).

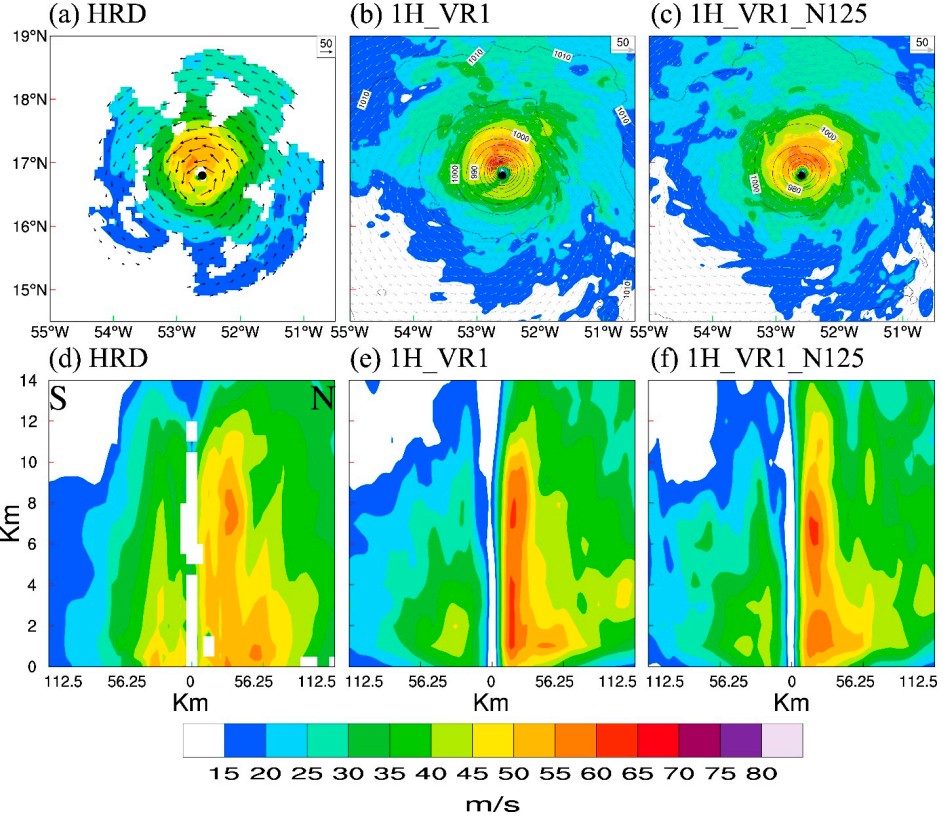

**Figure 9.** Same as Figure 3, except for (**b**,**e**) "1H_VR1" and (**c**,**f**) "1H_VR1_N125".

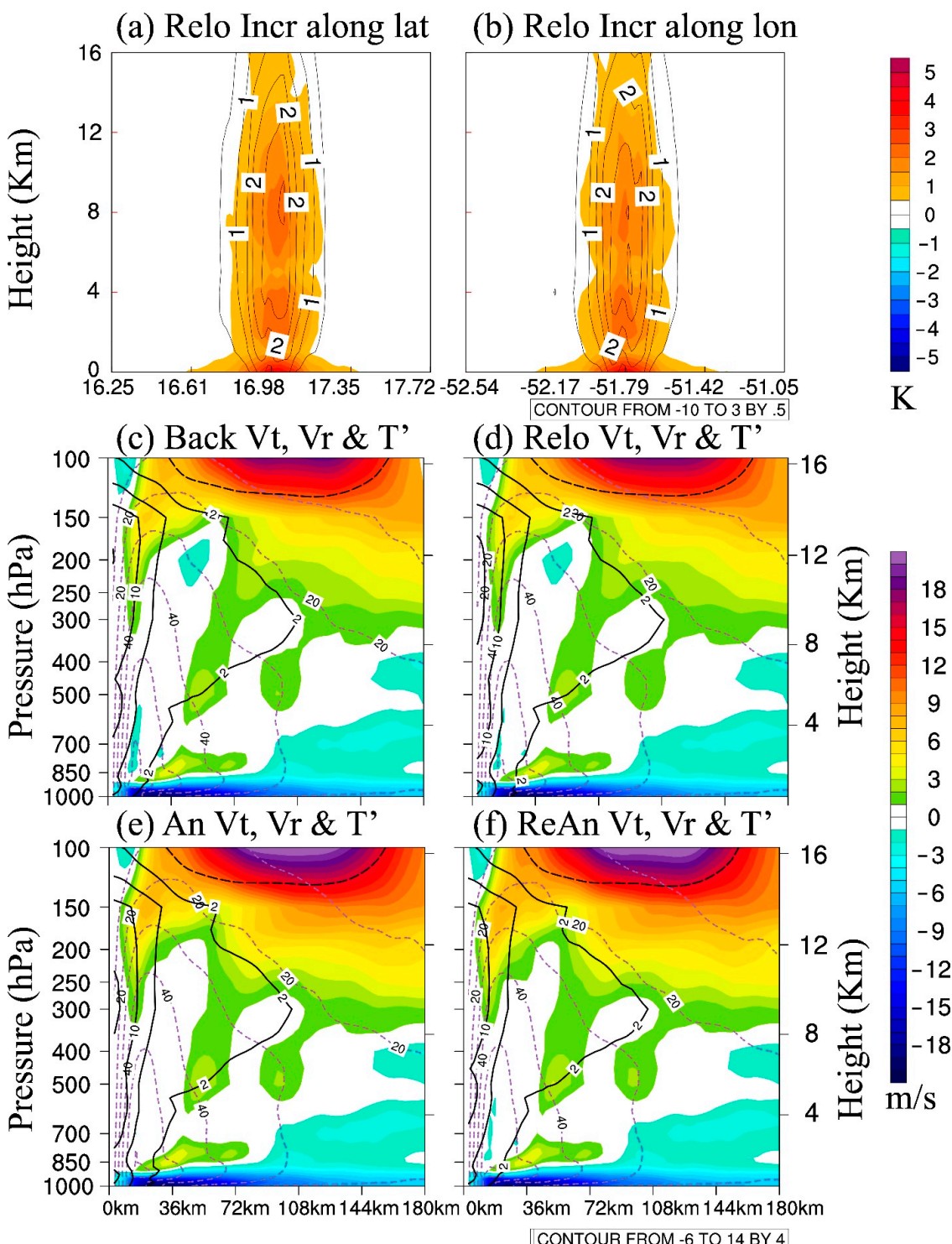

**Figure 10.** Temperature increment (shading) and pressure increment (contour) from VR for the (**a**) south-to-north and (**b**) west-to-east cross-sections valid at 07:00 UTC, 4 September 2017. (**c**–**f**) is the same as Figure 6, except for (**c**) Background storm, (**d**) VR storm, (**e**) DA analysis based on (**c**), and (**f**) DA analysis based on (**d**) valid at 07:00 UTC, 4 September 2017.

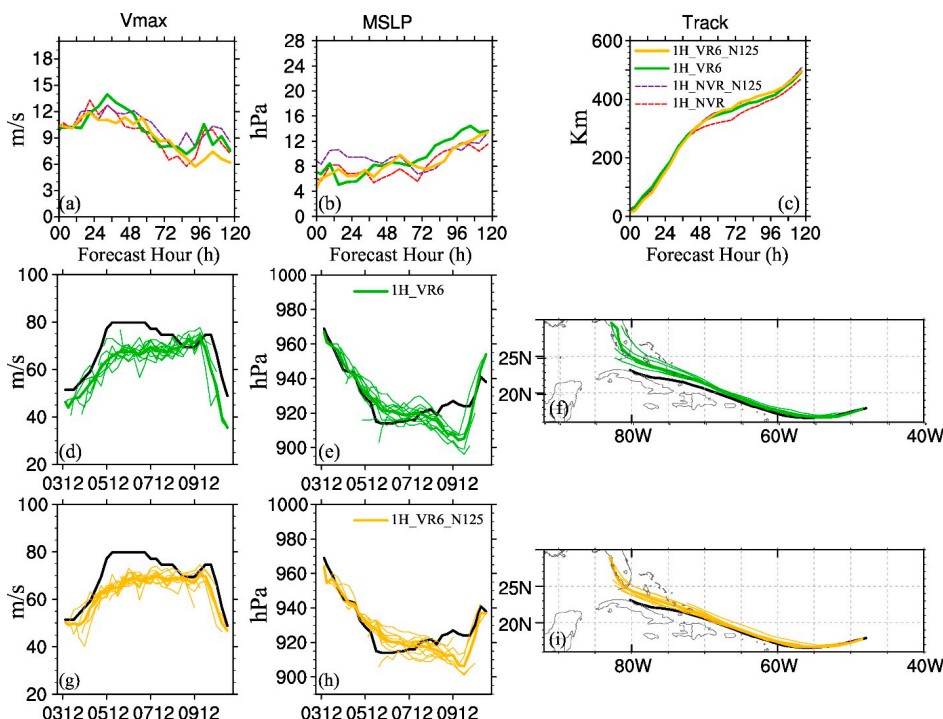

**Figure 11.** Same as Figure 2, except for "1H_VR6" (solid green), "1H_VR6_N125" (solid gold), "1H_NVR" (dashed red), and "1H_NVR_N125" (dashed purple).

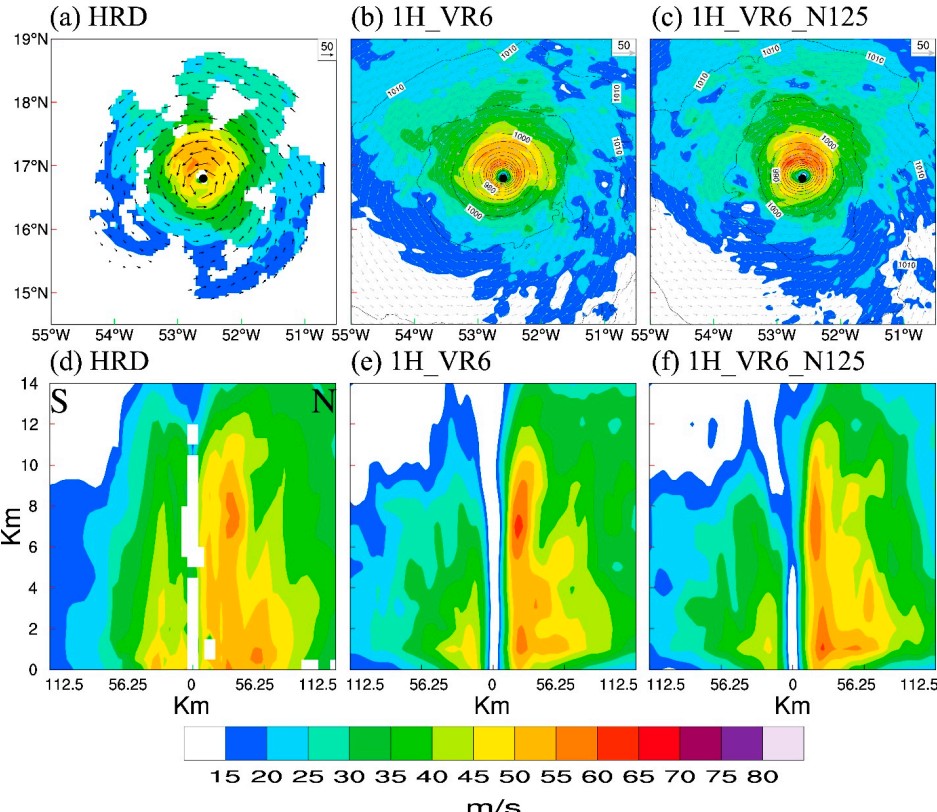

**Figure 12.** Same as Figure 3, except for (**b**,**e**) "1H_VR6" and (**c**,**f**) "1H_VR6_N125".

When comparing the "1H_VR6" experiment with the corresponding "6H_VR" experiment (e.g., dashed lines in Figure 13a–c), the hourly 3DEnVar experiments are not improving the intensity or track predictions upon the 6-hourly 3DEnVar experiments as

the NVR experiments, even though the spin-down issues are less frequent (Figure 11 vs. Figure 4). Nevertheless, the hourly 3DEnVar analysis produces a reasonably intensifying storm with better inner-core structures (Figure 12b,e vs. Figure 5b,e), and stronger and higher upper-level outflow (Figures 6 and 7) than the 6-hourly 3DEnVar analysis. While Lu and Wang (2019) [55] suggest that the lack of realistic model physics can be detrimental to better analysis, it requires further investigations in future work to better understand why the more reasonable analysis produced in "1H_VR6" is not improving the predictions upon "6H_VR".

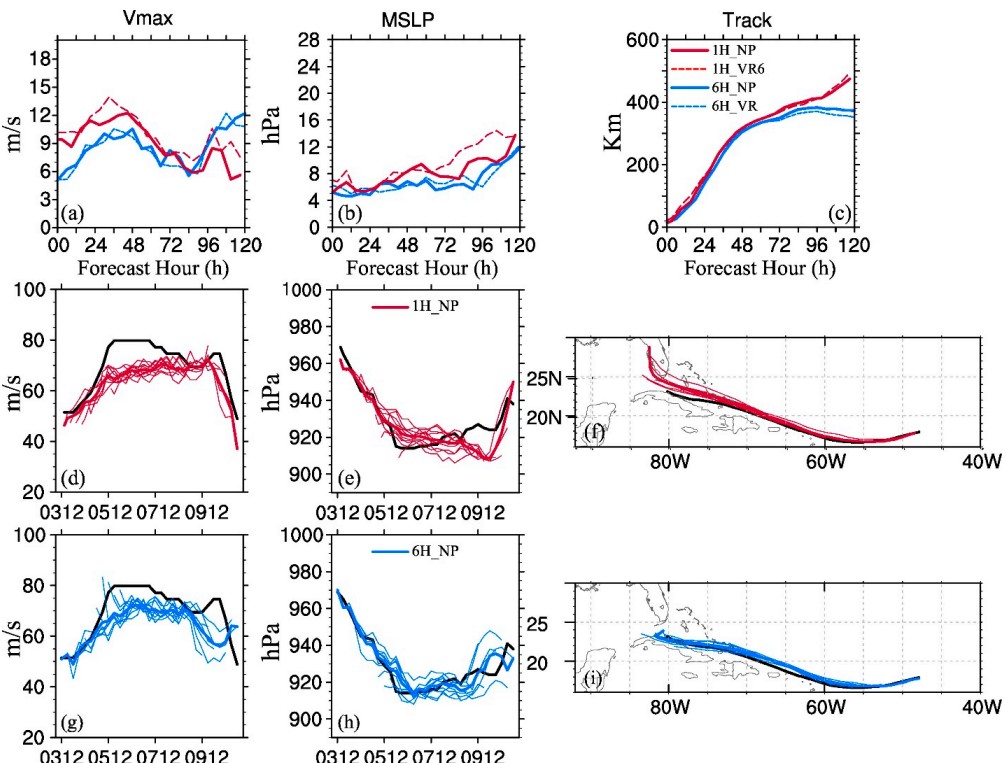

**Figure 13.** Same as Figure 2, except for "1H_NP" (solid red), "6H_NP" (solid turquoise), "1H_VR6" (dashed red), and "6H_VR" (dashed turquoise).

### 3.3. Impact of the Modified Error Profile

As stated in the Introduction, the default observation error profile used in the operational HWRF may not fit the enhanced AMVs dataset and needs to be updated. Therefore, this subsection investigates the impact of the updated error profile, as shown in Figure 1d, following Section 2.4.

Figure 1d shows that the updated error profile primarily increases the observation error above 200hPa. This increase in the upper level is primarily due to having more observations in this newer dataset to directly calculate the RMSE, given that there are more observations in regions where previously there were none due to data processing limitations (personal communication with Chris Velden and William Lewis, 2020). The profile change would primarily affect the storm's inner-core and near-core-vicinity regions, as well as deep convection regions, such as ones above the rainbands (not shown). Figure 13 shows that the updated error profile merely improves the predictions from the 6-hourly 3DEnVar experiments statistically, although "6H_NP" shows fewer spin-down issues and better MSLP trends during the weakening period. Figure 6c indicates that the secondary circulation in "6H_NP" is enhanced to be more consistent with the intensifying storm pattern. However, Figure 7c suggests that the strengthened outflow may not be high enough and is even producing fewer southerly components as compared to "6H_VR" in the observed southern wind regions near 57°W and 16.5–17.5°N around 100 hPa.

In comparison to the 6-hourly 3DEnVar experiments, the updated error profile benefits the hourly 3DEnVar more in the Vmax and MSLP predictions according to Figure 13. Figures 6 and 7 show that the differences between "1H_NP" and "1H_VR6" in the inner-core dynamical fields are small, but the upper-level warm core is slightly higher and warmer in "1H_NP". For an intensifying hurricane, it is more reasonable to see a strong and high upper-level outflow in agreement with its strong and warm upper-level warm core, which both indicate a stronger secondary circulation. Note that additional NP experiment based on "1H_NVR" shows comparable results with "1H_NP" and is therefore not shown.

## 4. Summary and Conclusions

To investigate the optimal DA configuration for the newly available inner-core-covered enhanced AMVs from CIMSS, multiple experiments are conducted in this study using the self-cycled GSI-based hybrid 3DEnVar DA system for HWRF during Hurricane Irma.

The investigations are first performed without VR. It is found that the hourly 3DEnVar can significantly outperform the 6-hourly 3DEnVar in such a scenario in almost all aspects, including the structure analysis and Vmax and MSLP predictions, except for the long-term track forecasts. Additionally, it is found that the assimilation of high-level inner-core AMVs can help improve the Vmax and MSLP predictions for both hourly and 6-hourly experiments. Especially for the hourly 3DEnVar, the additional assimilation of inner-core AMVs improves both Vmax, MSLP, and track predictions for almost all lead times.

Then, the experiments with VR show that VR benefits the 6-hourly 3DEnVar experiments the most. The assimilation of additional inner-core AMVs further improves the analysis and predictions in such a configuration. However, the VR can be detrimental to the hourly 3DEnVar DA configuration, especially for the MSLP predictions. Reducing VR frequency improves the intensity predictions while improving the structure analysis. However, the benefit of assimilating inner-core AMVs is still suppressed by the VR issue in such configurations. An intercomparison between the hourly and 6-hourly 3DEnVar experiments suggests that, although hourly 3DEnVar DA produces reasonable structure analyses, which are more consistent with an intensifying storm, it does not outperform the corresponding 6-hourly 3DEnVar DA when VR is performed every 6 h in both configurations.

Additional experiments with updated observation error profiles for the enhanced AMVs showed more improvements in the hourly 3DEnVar DA configuration than the 6-hourly 3DEnVar DA configuration.

The above results from the experiments indicate that:

1. The background location error is a key concern for a cycling 6-hourly 3DEnVar DA configuration. Consequently, there is a significant improvement when using VR to resolve the background location error issue before DA. Assimilating inner-core AMVs additionally improves the intensity predictions in the VR scenario.
2. The hourly 3DEnVar DA of the enhanced AMVs is less concerned by the background location error. It appears that in this Hurricane Irma case, the hourly assimilation of the enhanced AMVs, especially with the inner-core AMVs, is enough to correct the background storm location for a continuously cycling DA. The hourly 3DEnVar DA can thus easily outperform its 6-hourly counterparts when no VR is performed.
3. Since the location error is not the major concern for the hourly 3DEnVar, the current VR technique used in the operational HWRF does more harm than good to the hourly 3DEnVar DA with the artificial warm core issue. Reducing the frequency of VR can only reduce the negative impacts. As a result, there are no apparent prediction advantages of the hourly 3DEnVar over the 6-hourly 3DEnVar in the assimilation of the enhanced AMVs with VR.
4. The updated observation error profile can help improve the analysis and predictions for hourly 3DEnVar DA of the enhanced AMVs. The improvements from the corresponding 6-hourly 3DEnVar DA are tiny.

Overall, this study suggests that the current best DA configuration for the enhanced AMVs is to perform 6-hourly 3DEnVar with VR or to perform hourly 3DEnVar without VR

or only 6-hourly VR. The assimilation of inner-core AMVs with updated observation error profiles is found to be primarily helpful in hourly 3DEnVar DA.

As a preliminary study with only one case, this study is not intended for drawing a statistically solid conclusion but rather for giving us ideas on how to improve our utilization of the newly available enhanced AMV datasets. Further investigations with more cases and larger sample sizes are needed in future work. Additionally, better VR methods for hurricanes are worth exploring. More discussion and research on how to better utilize the improved DA to produce better intensity prediction are needed as well.

**Author Contributions:** Conceptualization, B.D., X.L. and X.W.; methodology, X.W.; software, X.L.; validation, B.D., X.L. and X.W.; formal analysis, X.L., B.D. and X.W.; investigation, B.D., X.L. and X.W.; resources, X.W.; data curation, X.W.; writing—original draft preparation, X.L.; writing—review and editing, X.L., B.D. and X.W.; visualization, X.L.; supervision, X.W.; project administration, X.W.; funding acquisition, X.W. All authors have read and agreed to the published version of the manuscript.

**Funding:** This research was funded by the National Oceanic and Atmospheric Administration, grant number NA16OAR4320115.

**Data Availability Statement:** All the observational data and global analysis used for the HWRF models and the model data produced during this study have been archived locally and are available upon request to the corresponding author.

**Acknowledgments:** The experiments were performed on the NOAA Jet supercomputer. Some post-processing and plotting were conducted on resources from the University of Oklahoma (OU) Supercomputing Center for Education and Research (OSCER). Some results and descriptions were included in the abstract of the authors' AMS conference presentation, the abstract of the second author's dissertation seminar, and progress reports to the funding agencies. We are grateful to Christopher Velden who provided the CIMSS AMV observations.

**Conflicts of Interest:** The authors declare no conflict of interest.

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
