# Peer review of "Improving the Assimilation of Enhanced Atmospheric Motion Vectors for Hurricane Intensity Predictions with HWRF"

_remotesensing, doi:10.3390/rs14092040_

Round 1
Reviewer 1 Report
See attached review.

Reviewer 2 Report
This study uses the Gridpoint Statistical Interpolation (GSI) -based three-dimensional (3D) hybrid Ensemble-Variational (EnVar) data assimilation system for the Hurricane Weather Research and Forecasting (HWRF) and shows the impact of the satellite-derived atmospheric motion vectors by the Cooperative Institute for Meteorological Satellite Studies (CIMSS) on the prediction of intensity, structure and their changes in the hurricane intensification phase. A lot of sensitivity experiments have been conducted in this paper, some of which are consistent with the best-track analysis and show the improvement. I have good impression for this manuscript because it is easy to read. I expect that this paper will be published. Before the publication, I would like to comment on a few points of concern
Major comments
・I have an impression that one of the major differences between AMV assimilation and the application of vortex relocation (VR) is the impact on the inner-core structure of the hurricane. In particular, the inner-core structure including the eye and eyewall becomes elliptical in the 6H_VR and 6H_VR_N125 experiments although the track and intensity predictions are comparable with the best-track analysis. What is the reason for the following points, unrealistic elliptical structure and improvement of the predictions? In addition, the authors should describe what kind of best-track dataset is used in this study.
・Except in the 6H_VR and 6H_VR_N125 experiments, most of the experiments that apply the VR seems to show a northward bias in the latter integration. What is the reason for this?
・I understand that the modification of the observation error profile for AMVs leads to the improvement of hurricane's intensity and track predictions. Unfortunately, any other results are not presented for the sensitivity experiments (1H_NP and 6H_NP). The results should be presented and be discussed in the revised manuscript.
・It is the outflow field in the upper troposphere that is affected by AMV assimilation. The authors should discuss the difference in the outflow field among sensitivity experiments. The wind field at 200 hPa altitude (lower than 150-hPa altitude) would be appropriate for presenting the difference.
Minor comments
・"DA" "3DEnVar" should be written down in the abstract.
・"3DEnVar" should be written down in the text.
・L155: 125 hPa. The unit should be unified in the manuscript. 
・L381: Please enter numbers within significant digits.
・Dashed lines are hard to see in all figures.
・Please indicate the abbreviations used in this paper as the Table.
・Please indicate the experiments conducted in this study as the Table.
Round 2
Reviewer 1 Report
Thank you for addressing my earlier suggestions - I think the manuscript is now acceptable for publication. Congratulations!
Author Response
Thanks! We appreciate your earlier comments, which help to further improve the manuscript.
Reviewer 2 Report
I would like to thank the authors for their appropriate response to my review comments in the initial review process. I do not think it is necessary to review the manuscript again, but I have some points of my concern shown below.
Minor comments
・L108 hurricane case
・(For examples) "201709031200 UTC" and others (many) should be written in the text as "2017/9/3 12:00 UTC" or "12 UTC on 3 September in 2017".
・In Table 1, L246-248 should be moved in section 2.1 if HURDAT2 best track data is used as Tropical Cyclone Vital minimum sea level pressure.
Author Response
We appreciate the comments from the reviewers, which help to further improve the manuscript.
The point-by-point response to all comments is provided below.
Minor comments
・L108 hurricane case
Re 1:
Thanks for pointing this out. The word has been modified in the revised manuscript at L146.
・(For examples) "201709031200 UTC" and others (many) should be written in the text as "2017/9/3 12:00 UTC" or "12 UTC on 3 September in 2017".
Re 2:
Thanks for pointing this out. The time format has been modified throughout the revised manuscript.
・In Table 1, L246-248 should be moved in section 2.1 if HURDAT2 best track data is used as Tropical Cyclone Vital minimum sea level pressure.
Re 3:
Thanks for considering this. But no, they are different. TCVital is the real-time estimated storm information. The HURDAT2 is a post-storm best track dataset determined by the National Hurricane Center, which makes use of all available observations.